# Estimation of the future prevalence of diabetes based on data from the Brazilian Study of Cardiovascular Risk Factors in Adolescents (ERICA)

**Barbara Pozzi Ottavio**[1]*, **Stéfani Sousa Borges**[2], **Márcia Gisele Santos da Costa**[3], **Maria Cristina Caetano Kuschnir**[4]

**1** Master's Degree in Health Technology Assessment, National Institute of Cardiology, Rio de Janeiro, Brazil and Ministry of Health, Brasília, Federal District, Brazil, **2** Faculty of Health Sciences, University of Brasília, Brasília, Federal District, Brazil, **3** National Institute of Cardiology, Rio de Janeiro, Brazil, **4** State University of Rio de Janeiro, Rio de Janeiro, Brazil

* barbara_pozzi@live.com

## Abstract

### Introduction

Diabetes is a significant public health issue due to its high prevalence and multifaceted consequences, impacting both the Brazilian Unified Health System and society. The disease adversely affects people's quality of life and elevates healthcare costs. In 2019, approximately 16 million Brazilians were diagnosed with diabetes mellitus. Understanding the future trends of this disease is crucial for planning effective preventive interventions. This paper aims to estimate the future prevalence type 2 diabetes mellitus on individuals from the Study of Cardiovascular Risk Factors in Adolescents (ERICA) sample, based on the progression of their cardiovascular risk factors.

### Method

A literature review was conducted to identify predictive models for type 2 diabetes mellitus that utilize cardiovascular risk factors assessed during adolescence to forecast diabetes risk in adulthood. A logistic regression model, grounded in the natural history of clinical variables derived from longitudinal studies, was applied to each individual to determine their risk of developing type 2 diabetes mellitus. Additionally, probabilistic and deterministic sensitivity analyses were performed, incorporating the minimum and maximum values of model parameters.

### Results

The predictive model estimated that 15.12% of individuals in the ERICA sample are likely to develop type 2 diabetes mellitus in adulthood, with a range of 1.1% to 28% based on sensitivity analyses. The parameters exerting the most significant influence

**Data availability statement:** This study utilized third-party data sources, including the ERICA Study, Vigitel, and the Bogalusa Heart Study. No additional individual-level data was generated or collected beyond what is included in these datasets. As we do not own these datasets, we are unable to publicly share the raw data. Researchers may refer to the respective data providers for further details on access conditions and availability. However, Iinterested researchers may request access through the respective data providers as per the following instructions.: • 1) ERICA Study: Data access requests can be submitted to the Brazilian Ministry of Health or the ERICA Study Coordination Team. Contact via projetoerica@gmail.com. Any researcher can request the data, explaining their research to justify the need for access. More details on ERICA's data availability can be found at http://www.erica.ufrj.br/index.php/o-erica/. • 2) Vigitel: The dataset is maintained by the Brazilian Ministry of Health and can be fully accessed by any researcher in their official portal: https://svs.aids.gov.br/download/Vigitel/ • 3) Bogalusa Heart Study: Researchers interested in accessing this dataset should direct inquiries to the Tulane Center for Cardiovascular Health, which manages study data access: https://medicine.tulane.edu/cardiology. For the present research, data from the published article was enough, so there was no need to request further information. No additional individual-level data was generated or collected beyond what is included in these datasets. Researchers may refer to the respective data providers for further details on access conditions and availability.

**Funding:** The author(s) received no specific funding for this work.

**Competing interests:** The authors have declared that no competing interests exist.

on these results included diastolic blood pressure and triglycerides, followed by LDL cholesterol and systolic blood pressure.

## Conclusion

The application of this predictive model to the Study of Cardiovascular Risk Factors in Adolescents (ERICA) sample indicates an estimated prevalence of 15.12% for T2DM over a 20.5-year follow-up period. Studies like this one provide valuable insights for designing targeted interventions to mitigate the progression of diabetes and its associated socioeconomic impacts.

---

## 1. Introduction

Diabetes Mellitus (DM) refers to a group of metabolic disorders characterized by high blood glucose levels resulting from defects in insulin production or action [1]. This chronic condition can lead to serious complications such as cardiovascular diseases, kidney failure, vision problems, and neuropathy [1]. The two most common types are Type 1 (T1DM), which is an autoimmune condition in which the insulin-producing cells are destroyed, and Type 2 (T2DM), which is marked by insulin resistance and is associated with obesity, aging, and a sedentary lifestyle. T2DM is the most prevalent form of diabetes, accounting for 90% of cases, while T1DM affects 5–10% of individuals. Less common forms of diabetes include gestational diabetes, monogenic diabetes, and secondary diabetes [2].

The prevalence of T2DM is rising rapidly with significant implications for global and national health systems. The International Diabetes Federation (IDF) reported that the number of people who have diabetes worldwide can grow from 463 million in 2019, to 700 million by 2045. In Brazil, the IDF projects that diabetes affected 16 million people in 2019, and that this number might reach 26 million by 2045 [3]. The rising prevalence also leads to an increase in national health expenditure. The Brazilian Ministry of Health annually conducts the Surveillance of Risk and Protection Factors for Chronic Diseases through Telephone Inquiry (Vigitel in Portuguese), research that maps the prevalence of chronic diseases and associated factors in the 26 city capitals from every state in addition to the Federal District. In 2023, 10.2% of respondents reported a diabetes diagnosis, almost double the 5.5% who reported it when Vigitel was launched in 2006 [4].

Diabetes-related costs in Brazil were calculated at BRL 52.3 billion in 2019 [3], approximately USD 13.27 billion at the 2019 exchange rate [3]. The Institute for Health Metrics and Evaluation (IHME) estimated that in 2017, 4.4% of the Brazilian population had diabetes. Among those aged 20 years of age and older, that figure increases to 6.2% with T2DM affecting 96% of this population [5,6]. Data on T2DM in children and adolescents are limited, as the condition typically presents later in life due to a combination of genetic and environmental factors.

Also according to the IHME, DM is among the leading causes of death, ranking fifth in 2021 and there has been a marked increase in deaths specifically attributed

to T2DM from 1990 (2.2% of the total, 17,532 individuals) to 2020 (4.38% of the total, 51,684 individuals). The years of life lost due to premature death (460,650.36 to 1,336,956.09) and years lived with disability (348,837.77 to 1,243,850.91) have also seen significant increases [7]. This scenario reflects the profound impact of diabetes on quality of life and the considerable burden of its complications. Such complications include microvascular issues like incipient diabetic retinopathy and neuropathy, as well as macrovascular problems such as cardiovascular disease. Consequently, Disability-Adjusted Life Years (DALY) – a metric that combines years of life lost and years lived with disability [8] – has risen dramatically from 991,854.76 to 2,580,807.00 [7].

Such a trend for increased prevalence, and financial impact, of T2DM is an important element for planning public policies. Literature provides varied projections, as the aforementioned data from the IDF and the IHME. The Ministry of Health also projects, based on Vigitel data, a growth in the prevalence of diabetes of about 9% (2.8% − 15.4%) for 2032 [4]. These data provide broad estimates based on a general population trend but do not specifically account for adolescent cardiovascular risk factors. Analyzing the T2DM tendency in a representative adolescent sample and considering their specific risk factors into the models could provide more precise estimates for the future generation.

The importance of T2DM has been pushing governments to action, leading the inclusion of initiatives and metrics in the Sustainable Development Goals (SDGs) and in the Strategic Action Plan for Addressing Chronic Diseases and Non-Communicable Diseases in Brazil 2021–2030. The SDG goal 3 aims to ensure healthy lives and promote well-being at all ages, with specific targets to reduce premature mortality from non-communicable diseases (NCDs) such as T2DM through prevention and treatment [9]. In parallel, Brazil's strategic plan admits the impact of T2DM on public health and on the economy, proposing actions to prevent and control the disease [10].

Given the impact of the disease, clinical guidelines are key to coordinating T2DM care. In Brazil, Clinical Protocol and Therapeutic Guidelines for T2DM set the diagnostic criteria and standard treatment for T2DM in the Brazilian public health system (SUS). Its latest version was published in 2024 by the Brazilian Ministry of Health [11]. The document establishes diagnostic tests and provides comprehensive treatment guidelines, including lifestyle modifications and pharmacological interventions, with metformin as the first-line treatment for T2DM.

Conducted between 2012 and 2014, the Study of Cardiovascular Risks in Adolescents (ERICA, acronym in Portuguese) is a national multicenter cross-sectional study. It aimed to estimate the prevalence of cardiovascular risk factors (CVRF) in approximately 75,000 students aged 12–17 from public and private schools in Brazilian cities with populations of 100,000 or more [12]. Although the ERICA sample does not represent all age groups, it does reflect the trends for a generation of students for which it is representative. Thus, estimates with ERICA's population can provide valuable input for public policy formulation and prioritizing health interventions that address health habits and conditions in this population, potentially mitigating risks for diabetes development.

Given the increasing impact of T2DM on public health, this study aims to estimate its future prevalence on ERICA's representative sample of adolescents by projecting the progression of their baseline CVRF and the resulting likelihood of developing T2DM.

## 2. Methods

A literature review was conducted in Embase and Medline databases in November 2023 to identify prediction models for DM that could contribute to its development. The search aimed to identify studies that utilized measures of CVRF collected during adolescence to predict the risk of diabetes in adulthood. The terms "risk factor," "adolescent," "diabetes," and "adult" were combined without restrictions by language or year. The search strategies are detailed in S1 File.

Evidence was sought focusing on studies that correlated childhood or adolescent CVRF with the development of T2DM in adulthood. Eligible studies were those that analyzed CVRF measured during childhood or adolescence, and also available for ERICA's sample, with the aim of predicting the development of T2DM in adulthood. Abstracts and articles selected for full reading were excluded if:

- Incorporated data not available in the ERICA study, such as genetic information, body fat mass, or family history of T2DM from members of the extended family.

- Referred to populations that are incompatible with the ERICA's, such as obese children or only with a family history of T2DM, adults, Chinese, Finns, or North American indigenous peoples.

- Included only one risk factor in the analysis, such as BMI or family history.

- Evaluated outcomes distinct from the desired outcome (development of T2DM), for example, applicability and definition of cardiovascular risk thresholds in childhood to predict T2DM in adulthood, or composite outcomes (fasting glucose, T2DM, and hypertension).

### 2.1. Model implementation

A model selected from the literature search was used to project the trend of adolescents in the ERICA sample in developing T2DM, utilizing an individual logistic regression that incorporated their characteristics. As per the inclusion criteria, the model incorporates risk factors that are available on ERICA. The model tracked individual patients with different demographic characteristics (age, sex, race) and clinical variables (such as blood pressure, BMI, and fasting glucose). Thus, a logistic regression was constructed with coefficients reflecting the natural history of clinical variables observed in a longitudinal study and applied to everyone in the sample.

The natural history was combined with the baseline data of ERICA's sample to i) estimate the individual probability of reaching or not the outcome in question, the development of T2DM, and then ii) the number of individuals in the sample who reached the outcome. To estimate the baseline risk of the ERICA population ($\beta_0$), data from Vigitel 2023 were considered, which indicate that 10.2% of the Brazilian population has diabetes, and that 90% of these cases are T2DM [4]. Thus, the intercept used should be ln (0.0918).

It was considered that an individual reached the outcome if they had a probability of having T2DM greater than or equal to 95%. The histograms with the frequency results of the probabilities are presented in S2 File. Additionally, probabilistic and deterministic sensitivity analyses were conducted using the maximum and minimum coefficients calculated from their corresponding confidence intervals. In the probabilistic analysis, all coefficients were varied simultaneously to their maximum and minimum to examine potential extreme scenarios. In the deterministic analysis, each coefficient was varied while keeping the others constant, to identify the most relevant elements for the results in the evaluated sample.

The application of the model and the other calculations were performed using R software (https://www.r-project.org/). The scripts used are found in S3 and S4 Files.

In a logistic regression, the natural log of the odds ratio is calculated, that is, the probability of an event occurring over the probability of it not occurring, based on the behavior of independent variables [13]. The equation below is used to calculate the natural log of the odds ratio.

$$logit\ (p) = \ln \left( \frac{p}{1-p} \right) = \beta_0 + \beta_1\ x_1 + .. + \beta_n\ x_n$$

The coefficient of a variable reflects how the odds ratio changes with an increase of one unit of that variable. Consequently, the probability of the event itself can be calculated using the following equation, which converts the natural logarithm into probability.

$$p = \left( \frac{e^{\beta_0 + \beta_1 x_1 + .. + \beta_n\ x_n}}{1 - e^{\beta_0 + \beta_1 x_1 + .. \beta_n x_n}} \right)$$

## 2.2. Characterization of T2DM

To understand the trajectory of individuals in the sample regarding T2DM, the model set thresholds characterizing the evolution of patients. These cut-off points served as references for the logistic regression applied to the present study and are aligned to those outlined in the Ministry of Health Clinical Protocol and Therapeutic Guidelines and international guidelines. For diabetes, the transition cut-off points are defined as follows: a fasting glucose level of ≥ 7 mmol/L and a glycated hemoglobin level greater than 6.5% (48 mmol/mol).

## 2.3. Predictive factors

To estimate the trajectory of each individual, it was necessary to establish which characteristics of each member of the sample could be included in the model, considering the natural history of T2DM. According to the International Diabetes Federation (IDF), the main risk factors that may lead to the development of T2DM include family history, overweight, unhealthy diet, physical inactivity, aging, hypertension, history of gestational diabetes, poor nutrition during pregnancy, glucose intolerance, and ethnicity. The American Diabetes Association (ADA) and the Brazilian Diabetes Society (SBD) [2,14] are more specific regarding aging, indicating that the risk increases after the age of 45, and include polycystic ovary syndrome in the list. The SBD further defines that there is a higher risk (i) for Afro-descendants, indigenous people, and Hispanics, (ii) when *acanthosis nigricans* is present, (iii) when HDL is below 35 mg/dL, and (iv) when triglycerides exceed 250 mg/dL.

In this context, the following indicators can be related to the predictive factors of T2DM and were mapped in the ERICA database to potentially compose the model in question:

- Socio-demographic data: age, race, and sex.

- Habits: physical activity.

- Family history: T2DM.

- Biochemical tests: total cholesterol, HDL cholesterol, LDL cholesterol, triglycerides, fasting glucose, glycated hemoglobin, fasting insulin.

- Anthropometry and blood pressure assessment: BMI, waist circumference, systolic blood pressure (SBP) and diastolic blood pressure (DBP).

Socio-demographic data and habits were collected through questionnaires completed by the students in the sample [12]. The form included sections on various topics, such as eating habits, alcohol consumption, and clinical history. Additionally, participants were required to indicate the presence of specific health conditions, including hypertension, diabetes, asthma, hypercholesterolemia, besides their body perception [12]. Students who reported having diabetes were excluded from the sample considered in this study, as they had already reached the evaluated outcome of developing T2DM or could not develop the disease due to having T1DM.

The family history was also collected through a questionnaire filled out by the parents of the adolescents, encompassing various characteristics such as socioeconomic data, weight, and height [12]. T2DM family history was considered the only relevant risk factor indicated in this questionnaire, as it has a direct impact on the possibility of an individual developing the disease [2,14,15].

The measurement of anthropometric parameters and blood pressure was performed by trained professionals following standardized procedures and using standardized stadiometers, precision scales, and anthropometric tape. For blood collection for the biochemical tests, participants were instructed to maintain a 12-hour overnight fast before the examination, which was performed in the morning. Blood samples were processed, with blood and serum separation within two hours after collection and maintained between 4°C and 10°C while transported to the study's sole laboratory, where they were stored at −80°C for future analyses [12].

Biochemical tests were conducted on the 40,732 students evaluated in ERICA who attended classes in the morning. Individuals who reported having diabetes were excluded, leaving only those who declared themselves not having the disease or who were unsure if they had it at the time of the ERICA assessment, totaling 35,738 adolescents.

The selected model was designed based on the evolution of individuals' CVRF – BMI, total cholesterol, HDL cholesterol, HDL cholesterol, triglycerides, SBP, and DBP – to explain the development of T2DM. The same CVRF are available for ERICA's individuals, which enabled the model to be applied to them.

## 2.4. Model validation

Internal validation was conducted to identify errors related to data entry and model programming. Biostatistics experts peer reviewed the methods and the scripts, ensuring methodological rigor. Experts with knowledge in epidemiology and on ERICA Study confirmed the appropriateness of applying the model to its sample.

In the absence of follow-up data on ERICA's sample clinical outcomes, it is not possible to assess whether individuals progressed to T2DM as predicted.

## 2.5. Limitations

There are limitations to the application of the selected model to ERICA's sample. First, the composition of both samples is similar, but their baseline age differs, which can impact on their clinical evolution. In addition, while the selected model was based in a study that followed cardiovascular risk factors also observed on ERICA Study, the location in which they were both conducted where different, and they started over 30 years apart. As the development of T2M relates to a person's habits, such discrepancies limit the comparability of the samples.

Certain potential confounding factors—such as socioeconomic status, dietary habits, genetic predisposition, and access to healthcare—were not incorporated into the model. While these factors are known to influence T2DM risk, they were not tracked or incorporated into the original model, so their exclusion was necessary to maintain data reliability and comparability with the reference study. This approach minimizes subjectivity and ensures that the projections are based solely on CVRF that have been consistently measured. Moreover, the model was developed to consider the full period of primary study, so it is limited to its duration and does not permit intermediate evaluations.

## 3. Results

### 3.1. Literature search

The literature search resulted in 2,024 articles, of which only one was eligible for inclusion. The selection process is illustrated in Fig 1.

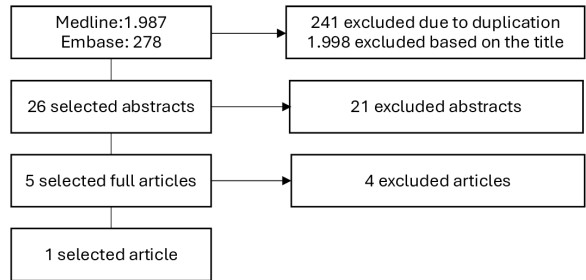

**Fig 1. Literature Search for the Predictive Model of Prevalence.**

The selected paper, "Variabilities in Childhood Cardiovascular Risk Factors and Incident Diabetes in Adulthood: The Bogalusa Heart Study", utilized data on CVRF to predict T2DM in adulthood. The model used data from a longitudinal study conducted in Bogalusa, Louisiana [16], by U.S. researchers to monitor CVRF in white and black individuals in a semi-rural setting since childhood (1973–2016). It aimed to estimate the risk of developing T2DM in adulthood by considering i) average levels of CVRF and ii) their variability among individuals. This approach sought to minimize confounding factors like age and comorbidities.

The research involved 20 data collection points—9 during childhood (ages 4–19) and 11 during adulthood (ages 20–58), occurring every 3–4 years. The mean follow-up period was 20.5 years. Individual monitoring included i) a structured questionnaire assessing family history of T2DM, medical history, medication use, smoking, and alcohol consumption, and ii) measurements of blood pressure, BMI, cholesterol (total, HDL, and LDL), triglycerides, and fasting glucose. The Bogalusa sample characteristics are presented in Table 1.

### 3.2. ERICA sample characteristics

ERICA's sample comprised 35,738 eligible individuals, whose data is detailed in Table 2, which presents the demographic and cardiovascular risk characteristics, stratified by sex. The average age of participants was approximately 14.6 years. Males displayed higher systolic blood pressure (114.38 mmHg) compared to females (108.40 mmHg), while diastolic blood pressure values were similar between the two groups (66.00 mmHg for males versus 66.78 mmHg for females) [17].

### 3.3. Logistic regression modeling

Based on the collected data of the Bogalusa longitudinal study, logistic regression was employed to assess the impact of both the average levels and variability of CVRF on the risk of developing T2DM. The study sample was divided into participants (1,718) and non-participants (1,455). The participant group included individuals with at least four measurements of CVRF recorded during childhood, and their data were utilized in the logistic regression models to estimate risk. The non-participant group served to adjust the models, and to test the risk estimates for internal validation.

Three non-linear growth curve models were employed to estimate how CVRF varied over the participants' lifetimes, considering: i) age, ii) age with a quadratic term for each risk factor, and iii) age with both quadratic and cubic terms for each risk factor. The authors used a random-effects model and selected the second model as the most appropriate, as it fitted all CVRF. Following this, logistic regressions were performed to assess the association of each cardiovascular risk factor with T2DM, identifying individuals with the disease based on fasting glucose levels of ≥ 126 mg/dL—the same threshold currently used.

The influence of sex and race on the results was tested, but the findings were similar across groups. Five models were applied to assess the influence of each cardiovascular risk factor, adjusting for various factors. Four indices were used to calculate interindividual variability: standard deviation (SD), coefficient of variation (CV), deviation from predicted age values (DEV), and residual standard deviation (RSD). Table 3 presents results of the five models and the four indices used for calculating variability.

The values are presented as Odds Ratios (95% CI) from logistic regression analyses. Model 1 was adjusted for mean age in childhood, sex, race, and adult age. Model 2 was adjusted for mean age in childhood, sex, race, adult age, parental history of diabetes, alcohol and smoking status, and the use of antihypertensive and lipid-lowering agents. Model 3 was adjusted for the variables in Model 2 and the respective mean values of CVRF in childhood. Model 4 was adjusted for the variables in Model 3 plus adult BMI and childhood BMI variability. Model 5 was adjusted for the variables in Model 4 plus parental education level and household income. As shown in Table 3, there was minimal variation in the results across the five models assessing the association between cardiovascular risk factor variability and the development of T2DM. Some risk attenuation occurred with additional adjustments, but the differences were small. Therefore, the simplest

**Table 1. Summary Data of the Bogalusa Study Sample Expressed as Averages by Sex and Race.**

| Race | White | | | | Black | | | |
|---|---|---|---|---|---|---|---|---|
| Variable | Males (n = 493) | SE | Female (n = 550) | SE | Male (n = 297) | SE | Female (n = 378) | SE |
| Age (Years) | 8.86 | 2.89 | 8.72 | 2.80 | 8.39 | 2.84 | 8.65 | 2.84 |
| BMI (Kg/m²) | 16.98 | 2.93 | 17.00 | 2.99 | 16.87 | 2.96 | 16.92 | 3.35 |
| Systolic blood pressure (mmHg) | 99.34 | 9.75 | 97.64 | 9.74 | 99.34 | 10.40 | 97.83 | 10.02 |
| Diastolic blood pressure (mmHg) | 50.52 | 9.10 | 50.44 | 9.18 | 52.20 | 9.15 | 51.25 | 9.44 |
| Total cholesterol (mg/dL) | 160.08 | 29.32 | 162.91 | 28.73 | 165.94 | 33.65 | 169.54 | 29.84 |
| HDL cholesterol (mg/dL) | 64.33 | 19.62 | 62.10 | 22.03 | 71.53 | 21.64 | 70.13 | 22.02 |
| LDL cholesterol (mg/dL) | 88.41 | 24.56 | 91.91 | 25.08 | 88.05 | 26.36 | 92.77 | 23.03 |
| Triglycerides (mg/dL) | 67.51 | 32.69 | 73.96 | 42.41 | 59.75 | 27.15 | 60.81 | 22.31 |
| Fasting plasma glucose (mg/dL) | 89.44 | 8.74 | 86.49 | 8.81 | 86.87 | 9.92 | 84.67 | 9.59 |

SE = standard error; BMI = body mass index.

Source: Variabilities in Childhood Cardiovascular Risk Factors and Incident Diabetes in Adulthood: The Bogalusa Heart Study [16].

**Table 2. Summary Data of the Cardiovascular Risk in Adolescents Study (ERICA) Sample Expressed as Means by Sex and After Weighting.**

| Variable | Female | SE | Male | SE |
|---|---|---|---|---|
| Age in years | 14.63 | 0.00 | 14.60 | 0.00 |
| BMI (kg/m² recalculated and adjusted) | 21.58 | 0.09 | 21.18 | 0.09 |
| Systolic blood pressure (mmHg) | 108.40 | 0.26 | 114.38 | 0.28 |
| Diastolic blood pressure (mmHg) | 66.78 | 0.20 | 66.00 | 0.20 |
| Total cholesterol (mg/dL) | 152.57 | 0.65 | 143.37 | 0.59 |
| HDL cholesterol (mg/dL) | 49.66 | 0.36 | 44.92 | 0.30 |
| LDL cholesterol (mg/dL) | 87.10 | 0.46 | 83.20 | 0.56 |
| Triglycerides (mg/dL) | 79.05 | 0.73 | 76.20 | 0.88 |

SE = standard error, BMI = body mass index, HDL = High-Density Lipoprotein, LDL = Low-Density Lipoprotein. Source: Prepared by the authors based on data from ERICA.

model—adjusted only for mean childhood age, sex, race, and mean adult age—was selected for this study. Similarly, since the four methods for analyzing variability produced comparable results, the SD was chosen as the reference for the ERICA sample due to its widespread use and straightforward interpretation.

The results are expressed as odds ratios (OR) from the logistic regressions, indicating the impact of a one-unit change in each independent variable on the probability of developing T2DM. To apply these coefficients to the ERICA sample, the values were converted to a logarithmic scale, as detailed in Table 4.

## 4. Results

### 4.1. Prevalence of T2DM

The model was applied utilizing its average coefficients on the 35,738 students from ERICA—individuals who indicated either not having or being uncertain about having DM—minus 119 whose data were unavailable. It was estimated that 5,386 individuals may develop the T2DM. This projection corresponds to a prevalence of 15.12%. The temporal horizon for this model is 20.5 years, reflecting the follow-up duration of the Bogalusa sample. This suggests that individuals in the ERICA cohort are likely to develop diabetes between the ages of 32 and 37, specifically between 2032 and 2034.

**Table 3. Association between Variabilities in Childhood Cardiovascular Risk Factors and Adult Type 2 Diabetes.**

| | Model 1 | Model 2 | Model 3 | Model 4 | Model 5 |
|---|---|---|---|---|---|
| **Body mass index** | | | | | |
| SD | 1.55 (1.32–1.81) | 1.48 (1.26–1.74) | 1.31 (1.06–1.61) | — | 1.30 (1.05–1.61) |
| CV | 1.50 (1.26–1.80) | 1.47 (1.22–1.76) | — | — | 1.46 (1.21–1.76) |
| DEV | 1.42 (1.24–1.63) | 1.34 (1.16–1.55) | 1.30 (1.02–1.66) | — | 1.29 (1.02–1.66) |
| RSD | 1.43 (1.24–1.63) | 1.35 (1.17–1.56) | 1.30 (1.01–1.66) | — | 1.29 (1.01–1.67) |
| **HDL cholesterol** | | | | | |
| SD | 1.37 (1.18–1.59) | 1.37 (1.17–1.60) | 1.39 (1.19–1.63) | 1.33 (1.12–1.57) | 1.31 (1.11–1.56) |
| CV | 1.39 (1.21–1.60) | 1.33 (1.14–1.55) | — | 1.23 (1.05–1.44) | 1.15 (0.95–1.39) |
| DEV | 1.23 (1.05–1.45) | 1.18 (1.01–1.40) | 1.17 (1.00–1.40) | 1.14 (1.00–1.47) | 1.13 (0.95–1.43) |
| RSD | 1.27 (1.08–1.49) | 1.23 (1.04–1.45) | 1.23 (1.04–1.46) | 1.19 (1.01–1.42) | 1.17 (1.01–1.40) |
| **Systolic blood pressure** | | | | | |
| SD | 0.94 (0.78–1.14) | 0.90 (0.74–1.10) | 0.89 (0.72–1.08) | 0.89 (0.72–1.11) | 0.90 (0.73–1.11) |
| CV | 0.89 (0.74–1.08) | 0.87 (0.71–1.06) | — | 0.89 (0.72–1.11) | 0.89 (0.72–1.11) |
| DEV | 1.22 (1.04–1.44) | 1.16 (0.98–1.37) | 1.09 (0.91–1.31) | 1.08 (0.90–1.31) | 1.08 (0.89–1.31) |
| RSD | 1.19 (1.00–1.41) | 1.13 (0.95–1.34) | 1.06 (0.88–1.27) | 1.04 (0.86–1.25) | 1.03 (0.85–1.25) |
| **Diastolic blood pressure** | | | | | |
| SD | 1.15 (0.97–1.36) | 1.17 (0.98–1.40) | 1.17 (0.98–1.40) | 1.18 (0.98–1.41) | 1.17 (0.97–1.40) |
| CV | 1.12 (0.94–1.33) | 1.17 (0.98–1.41) | — | 1.21 (1.01–1.46) | 1.19 (0.99–1.44) |
| DEV | 1.10 (0.93–1.30) | 1.08 (0.90–1.29) | 1.08 (0.90–1.29) | 1.05 (0.87–1.26) | 1.04 (0.87–1.26) |
| RSD | 1.09 (0.92–1.29) | 1.08 (0.90–1.29) | 1.08 (0.90–1.29) | 1.04 (0.86–1.25) | 1.03 (0.86–1.24) |
| **Total cholesterol** | | | | | |
| SD | 1.16 (0.99–1.36) | 1.11 (0.93–1.32) | 1.10 (0.92–1.32) | 1.04 (0.96–1.26) | 1.03 (0.85–1.25) |
| CV | 1.14 (0.97–1.35) | 1.14 (0.94–1.32) | — | 1.05 (0.88–1.26) | 1.04 (0.87–1.24) |
| DEV | 1.23 (1.06–1.42) | 1.17 (1.01–1.36) | 1.18 (0.99–1.39) | 1.18 (0.99–1.40) | 1.17 (0.98–1.39) |
| RSD | 1.24 (1.07–1.44) | 1.18 (1.01–1.38) | 1.20 (1.01–1.42) | 1.19 (0.99–1.42) | 1.18 (098–1.41) |
| **Triglycerides** | | | | | |
| SD | 1.00 (0.84–1.20) | 0.99 (0.82–1.19) | 0.94 (0.79–1.13) | 0.89 (0.74–1.08) | 0.90 (0.74–1.09) |
| CV | 0.95 (0.79–1.15) | 0.95 (0.79–1.14) | — | 0.90 (0.74–1.09) | 0.91 (0.75–1.10) |
| DEV | 1.15 (0.98–1.36) | 1.11 (0.93–1.31) | 0.99 (0.82–1.20) | 0.96 (0.79–1.17) | 0.97 (0.79–1.19) |
| RSD | 1.16 (0.98–1.36) | 1.12 (0.95–1.33) | 1.01 (0.83–1.22) | 0.97 (0.80–1.18) | 0.97 (0.80–1.19) |
| **LDL cholesterol** | | | | | |
| SD | 1.30 (1.12–1.52) | 1.24 (1.05–1.46) | 1.22 (1.02–1.47) | 1.18 (0.98–1.43) | 1.18 (0.97–1.43) |
| CV | 1.25 (1.07–1.47) | 1.24 (1.05–1.47) | — | 1.21 (0.99–1.44) | 1.20 (1.01–1.43) |
| DEV | 1.23 (1.06–1.42) | 1.15 (0.98–1.34) | 1.11 (0.93–1.33) | 1.14 (0.95–1.37) | 1.14 (0.95–1.37) |
| RSD | 1.25 (1.09–1.46) | 1.18 (1.01–1.37) | 1.15 (0.96–1.39) | 1.18 (0.97–1.43) | 1.18 (0.97–1.42) |

SD = standard deviation; CV = coefficient of variation; DEV = deviation from age-predicted values; RSD = residual standard deviation. Source: Variabilities in childhood cardiovascular risk factors and incident diabetes in adulthood: The Bogalusa heart study [16].

## 4.2. Sensitivity analyses

To assess how these estimates might vary, probabilistic and deterministic sensitivity analyses were performed. Coefficients had been statistically estimated according to the clinical evolution of the Bogalusa sample.

The findings from the probabilistic sensitivity analysis, presented in Table 5, indicate that the percentage of individuals who may develop T2DM within the sample could range from 1.1% to 28%.

**Table 4. Coefficient Conversion.**

| | BMI | LDL cho-lesterol | Systolic blood pressure | Diastolic blood pressure | Total cho-lesterol | Tri-glycerides | LDL cho-lesterol |
|---|---|---|---|---|---|---|---|
| Average SD of OR | 1.55 | 1.37 | 0.94 | 1.15 | 1.16 | 1 | 1,3 |
| Average SD Coefficient | 0.438 | 0.315 | −0.062 | 0.140 | 0.148 | 0.000 | 0.262 |
| Minimum Limit SD of OR (95% CI) | 1.320 | 1.180 | 0.780 | 0.970 | 0.990 | 0.840 | 1.120 |
| Minimum Limit Coefficient | 0.278 | 0.166 | −0.248 | −0.030 | −0.010 | −0.174 | 0.113 |
| Maximum Limit SD of OR (95% CI) | 1.810 | 1.590 | 1.140 | 1.360 | 1.360 | 1.120 | 1.520 |
| Maximum Limit Coefficient | 0.593 | 0.464 | 0.131 | 0.307 | 0.307 | 0.113 | 0.419 |

SD = Standard Deviation; OR = Odds Ratio; BMI = Body Mass Index.

Source: Variabilities in Cardiovascular Risk in Childhood and Incidence of Diabetes in Adulthood: The Bogalusa heart study [16].

**Table 5. Results of the probabilistic sensitivity analysis.**

| | Minimum coefficient | Average coefficient | Maximum coefficient |
|---|---|---|---|
| Number of Individuals with T2DM | 374 | 5,386 | 9,960 |
| Percentage of Individuals with T2DM | 1.1% | 15.1% | 28.0% |

Table 6 displays the results of the deterministic sensitivity analysis. The parameters exerting the most significant influence on the results are diastolic blood pressure and triglycerides, followed by LDL cholesterol and systolic blood pressure. The variation in the BMI coefficient had a relatively minor effect on the results, while an increase in the HDL coefficient led to a decrease in the estimated number of individuals with T2DM.

## 4.3. Regional prevalence estimates

The results were distributed by state, reflecting the average coefficients for T2DM prevalence in various regions. The highest prevalence rates among individuals in the ERICA study were observed in the following regions: the Federal District (DF) in the Central-West region at 23.7%, Santa Catarina (SC) in the South region at 22.9%, Minas Gerais (MG) in the Southeast region at 20.6%; Rio Grande do Sul (RS) and Paraná (PR) in the South region at 19.7% and 18.3%, respectively; and Rio de Janeiro (RJ) in the Southeast region at 18%. In contrast, the states with the lowest estimated prevalence in the ERICA sample included four from the North region – Rondônia (RO) at 8%, Amazonas (AM) at 9.3%, Amapá (AP) at 9.7%, and Tocantins (TO) at 10.86% - besides one in the Central-West – Mato Grosso (MT) at 9.7%. The estimated prevalence by state is illustrated in Fig 2 and detailed in Table 7.

**Table 6. Results of the deterministic sensitivity analysis.**

| Parameter | Minimum coefficient | | Maximum coefficient | | Difference in results |
|---|---|---|---|---|---|
| | In (coefficient) | Individuals with T2DM | In (coefficient) | Individuals with T2DM | |
| BMI | 0.278 | 4,850 | 0.593 | 5,888 | 1038 |
| HDL cholesterol | 0.166 | 6,919 | 0.464 | 4,455 | −2464 |
| Systolic blood pressure | −0.248 | 4,162 | 0.131 | 6,808 | 2646 |
| Diastolic blood pressure | −0.030 | 3,865 | 0.307 | 7,097 | 3232 |
| Total cholesterol | −0.010 | 3,877 | 0.307 | 6,155 | 2278 |
| Triglycerides | −0.174 | 4,184 | 0.593 | 7,507 | 3323 |
| LDL Cholesterol | 0.113 | 3,769 | 0.392 | 6,480 | 2711 |

BMI = Body Mass Index.

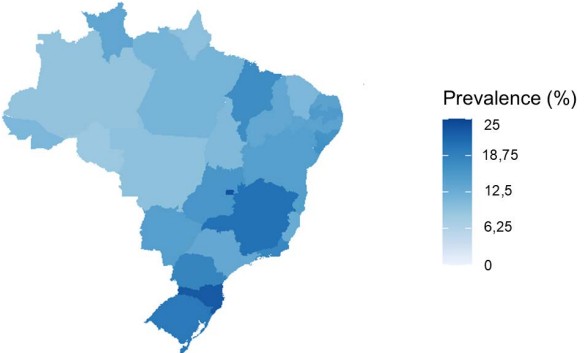

Prevalence (%)

25
18,75
12,5
6,25
0

**Fig 2. Estimated future prevalence of T2DM by state in the ERICA sample.** *This figure was created by the authors and is published under the Creative Commons Attribution 4.0 International (CC BY 4.0) license.

**Table 7. Estimated future prevalence of T2DM by state in the ERICA sample.**

| State | Sample by state | Estimated number of with T2Dm | Prevalence |
|---|---|---|---|
| AC | 810 | 90 | 11.11% |
| AL | 1,095 | 179 | 16.35% |
| AM | 1,290 | 120 | 9.30% |
| AP | 531 | 52 | 9.79% |
| BA | 1,100 | 162 | 14.73% |
| CE | 1,589 | 176 | 11.08% |
| DF | 1,160 | 275 | 23.71% |
| ES | 848 | 108 | 12.74% |
| GO | 1,875 | 300 | 16.00% |
| MA | 1,379 | 238 | 17.26% |
| MG | 2,072 | 428 | 20.66% |
| MS | 745 | 110 | 14.77% |
| MT | 1,442 | 140 | 9.71% |
| PA | 2,559 | 296 | 11.57% |
| PB | 1,062 | 164 | 15.44% |
| PE | 2,129 | 301 | 14.14% |
| PI | 888 | 120 | 13.51% |
| PR | 2,009 | 368 | 18.32% |
| RJ | 2,204 | 397 | 18.01% |
| RN | 795 | 113 | 14.21% |
| RO | 673 | 58 | 8.62% |
| RR | 381 | 52 | 13.65% |
| RS | 1,624 | 321 | 19.77% |
| SC | 867 | 199 | 22.95% |
| SE | 970 | 163 | 16.80% |
| SP | 2,988 | 398 | 13.32% |
| TO | 534 | 58 | 10.86% |

Furthermore, the regional distribution of T2DM prevalence is noteworthy, with significantly higher rates observed in the Southeast and South regions, as highlighted in Table 8.

## 5. Discussion

The application of the logistic regression model to the ERICA sample provides valuable insights on the potential future impacts of worsening cardiovascular risk indicators on the population over the medium term. We estimate a prevalence of 15.12% for T2DM among individuals in the sample over a period of 20.5 years, reflecting the increasing prevalence of the disease in Brazil. This estimate exceeds the International Diabetes Federation (IDF) projections, aligning more closely with the upper limits forcasted by the Brazilian Ministry of Health. While the IDF estimates that 10.2% of the population will have diabetes by 2030, potentially rising to 10.9% by 2045, the Brazilian Ministry of Health anticipates a trend of 9.1% for 2032, with a range from 2.8% to 15.4%. Notably, between 2019 and 2023, the percentage of diabetes among adults in Brazil rose from 7.4% to 10.2%, reflecting an annual increase of approximately 0.5 percentage points [4].

In comparison, the analytical sample from the Bogalusa model included 1,718 individuals, of whom 133 developed diabetes during the follow-up period, resulting in a prevalence of 7.7% - approximately half of the rate found in the ERICA sample. This discrepancy may be attributed to differences in baseline characteristics between the two populations, particularly their average ages at study enrollment. Participants in the ERICA study were aged 12–17 years (mean age of 14.6 years), whereas those in the Bogalusa study had an average age of around 8 years. The variation may also reflect differences in developmental experiences, dietary habits, cultural practices, and other environmental factors influencing T2DM trajectories across these cohorts, especially since they are from different countries and not contemporary. Despite these differences, the prevalence reported in the ERICA sample is not far from the 8.9% prevalence of diabetes in Brazil recorded in 2016 [18].

Study estimates that the prevalence of pre-diabetes among ERICA students exceeded 20% at the moment data was collected, indicating a concerning trend towards an increased risk of developing T2DM [19]. Such results align to the present article's conclusion that about 15% of the sample may develop T2DM 20 years afterwards. In addition, over 3.3% of the students had already developed T2DM, underscoring the association at a very young age between unhealthy eating habits, physical inactivity, and other modifiable risk factors over the long term, along with the influence of aging [19].

Similar discrepancies in time and geography are observed in the study conducted by the International Consortium of Child Cardiovascular Risk Cohorts (i3C) [20]. In addition to the Bogalusa study, the i3C analysis included five other cohorts—four from the U.S. and one from Finland—whose baseline measures were collected from the 1970s to the 1990s. Among a total of 6,738 participants, 6.5% developed T2DM, a figure lower than the 15.1% prevalence found in this study but comparable to the 7.7% reported in the Bogalusa study. These variations may be attributed to changes in dietary habits and increased sedentary behavior over time, given that the ERICA survey was conducted more recently.

The deterministic sensitivity analysis revealed varying influences of the assessed parameters on the results. Notably, the coefficient for BMI—the most significant parameter—exhibited minimal impact on the overall findings. Conversely, an

**Table 8. Prevalence of T2DM in 2023 and estimated prevalence of T2DM in the ERICA sample.**

| Region | Prevalence of T2DM in Brazil in 2023 (23) | Future prevalence of T2DM in the ERICA sample |
|---|---|---|
| Central-West | 10.5% | 15.8% |
| Northeast | 9.4% | 14.7% |
| North | 7.3% | 10.7% |
| Southeast | 11.3% | 16.4% |
| South | 10.3% | 19.7% |

**Source:** Vigitel 2023 [4].

increase in HDL cholesterol resulted in a decrease in the estimated number of individuals developing T2DM. Although triglycerides had a lower overall weight compared to other parameters, they demonstrated substantial variability, likely due to diverse baseline values within the assessed population. Additionally, diastolic blood pressure and LDL cholesterol were found to significantly impact the outcomes.

Probabilistic sensitivity analysis indicates the prevalence estimation could range from 1.1% to 28%, depending on the weight applied to cardiovascular risk factors included in the model. However, as cardiovascular disease and diabetes are intertwined, and risk factors are common to both [1,2,4], it is unlikely for all CVRF to have the lower limit weight on the development of T2DM. Having some sort of interval is still useful given the limitations of the analysis, the natural heterogeneity of the population, and the multifactorial influences on the development of the disease, as they inform decision-makers on the possible different scenarios.

Considering the economic implications of T2DM, estimated costs indicate that both direct and indirect costs are already considerable. In 2014, costs reached USD 15 billion, corresponding to a prevalence rate of 6.2%. If the estimated prevalence of 15.1% is confirmed, the country could potentially see 30.6 million individuals with T2DM, as projected by the 2022 Demographic Census, resulting in a significant increase in healthcare costs [21].

While these results highlight an upward trend in T2DM prevalence, it is essential to acknowledge its limitations. First, the trajectory was modeled based on a cohort from another country, which may not accurately represent the same cardiovascular risk factors for the Brazilian population. Differences in genetic, socioeconomic, dietary patterns, and levels of physical activity could lead to varying results between cohorts, especially given the relationship between habits and the development of T2DM. Furthermore, the model was developed with data from Bogalusa's cohort, which was monitored for an average of 20.5 years. As a consequence, it is challenging to extrapolate these findings to larger time-frames. Given that age is a known risk factor for T2DM and considering the mean age of ERICA participants at baseline, the projections may underestimate the disease incidence within this sample.

Additionally, we were unable to ascertain whether participants in the Bogalusa study underwent any interventions—such as medication use or dietary changes—in cases of increased cardiovascular risk. It is also not possible to understand if there have been any interventions in Brazil during the growth of ERICA's individuals that could have changed their probability of developing T2DM. Also, the model exclusively included clinical variables, while there is extensive literature on how social determinants of health influence the progression of T2DM, such as income, education, housing, access to food and healthcare. Differences in the Bogalusa and ERICA samples regarding these determinants are not quantifiable but might also limit the interpretation of results. Future studies could incorporated socio-economic factors in addition to the clinical ones, to enable a better understanding of the Brazilian reality [22].

Finally, the baseline risk applied to the ERICA sample was derived from the Vigitel telephone survey conducted annually by the Ministry of Health. While this sample may not perfectly represent the broader Brazilian population due to factors such as self-reported data and geographic restrictions, it offers valuable insights.

The findings of this study indicate a rising prevalence of T2DM in Brazil, driven by identified risk factors present during adolescence. This underscores the urgent need for public health interventions from childhood to prevent the progression of diabetes. Initiatives to ensure access to healthy foods, implement policies to reduce the consumption of ultra-processed foods and alcohol, promote physical activity at all ages, and combat smoking are essential actions that should be prioritized to enhance public health and mitigate the advancement of this disease in the country.

By leveraging data from the ERICA study and employing logistic regression models, we estimated the potential risk of each individual for developing T2DM, providing valuable insights into the landscape of risk factors and diabetes prevalence. These strategies should focus on targeted interventions for at-risk groups, aiming to mitigate the development of T2DM and its associated complications. Ultimately, the findings will not only enhance our understanding of diabetes epidemiology among young people but also serve as a foundational tool for the development of informed public policies,

enabling health authorities to implement preventive measures and allocate resources more effectively to improve the quality of life for this population.

## 6. Conclusion

This study successfully applied a predictive model developed in the U.S. to estimate the future prevalence of T2DM in Brazilian adolescents from the ERICA sample. The analysis estimated that approximately 15.12% of individuals in the sample could develop T2DM over a 20.5-year period, a figure that, while consistent with the rising trends of the disease in Brazil, is subject to significant variation depending on CVRF. The deterministic analysis reveals that higher HDL cholesterol levels are associated with a lower probability of developing T2DM, while increases in diastolic blood pressure and triglycerides, along with elevated LDL cholesterol and systolic blood pressure, are linked to a higher likelihood of the disease.

While the results provide valuable insights for formulating public health policies aimed at preventing and controlling T2DM in Brazil, it is essential to recognize the inherent limitations of using a foreign model in a population with different socioeconomic and cultural characteristics. Future studies should aim to develop or adjust predictive models based on national cohorts, which would allow for greater precision and suitability for the Brazilian context.

Interventions focused on factors such as blood pressure control and the reduction of triglyceride levels may be effective strategies for mitigating the risk of developing T2DM. Furthermore, public health policies that promote lifestyle changes from adolescence—such as encouraging healthy eating habits and increasing physical activity—could play a crucial role in reducing the future burden of the disease.

## Supporting information

**S1 File. Search strategies for Medline and Embase were applied without restrictions on period or language, limiting the searches to title and abstract.**
(PDF)

**S2 File. Histograms showing the frequency of the probabilities of developing type 2 diabetes in the ERICA sample for the maximum, mean, and minimum coefficients, considering the confidence intervals.**
(TIF)

**S3 File. Script for adjustments to the ERICA sample, including weighting according to instructions and cleaning the dataset to remove individuals not eligible for the present study.**
(PDF)

**S4 File. Script for applying the Bogalusa model to the ERICA database and sensitivity analyses.**
(PDF)

**S5 File. List of studies excluded after full-text review.**
(DOCX)

## Author contributions

**Conceptualization:** Barbara Pozzi Ottavio, Márcia Gisele Santos da Costa.

**Data curation:** Barbara Pozzi Ottavio.

**Formal analysis:** Barbara Pozzi Ottavio.

**Investigation:** Barbara Pozzi Ottavio.

**Methodology:** Barbara Pozzi Ottavio.

**Project administration:** Barbara Pozzi Ottavio.

**Supervision:** Márcia Gisele Santos da Costa, Maria Cristina Caetano Kuschnir.

**Writing – original draft:** Barbara Pozzi Ottavio, Stéfani Sousa Borges.

**Writing – review & editing:** Barbara Pozzi Ottavio, Márcia Gisele Santos da Costa, Maria Cristina Caetano Kuschnir.

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
