## [Decision Letter · Decision Letter 0]

10 Mar 2025

PONE-D-24-44409Estimation of the future prevalence of diabetes based on data from the Brazilian Study of Cardiovascular Risk Factors in Adolescents (ERICA).PLOS ONE

Dear Dr. Pozzi Ottavio,

Thank you for submitting your manuscript to PLOS ONE. After careful consideration, we feel that it has merit but does not fully meet PLOS ONE’s publication criteria as it currently stands. Therefore, we invite you to submit a revised version of the manuscript that addresses the points raised during the review process.

The manuscript has been evaluated by three reviewers, however please note that the comments from reviewer 1 refer to a different manuscript and can be disregarded. The reviewer comments are available below.

The reviewers have raised a number of concerns that need attention. In particular, they request additional information on methodological aspects of the study and further discussion

Could you please revise the manuscript to carefully address the concerns raised by reviewers 2 and 3?

We look forward to receiving your revised manuscript.

Kind regards,

Helen Howard

Staff Editor

PLOS ONE

Journal Requirements:

3. For studies involving third-party data, we encourage authors to share any data specific to their analyses that they can legally distribute. PLOS recognizes, however, that authors may be using third-party data they do not have the rights to share. When third-party data cannot be publicly shared, authors must provide all information necessary for interested researchers to apply to gain access to the data. (https://journals.plos.org/plosone/s/data-availability#loc-acceptable-data-access-restrictions) 

5. We note that Figure 2 in your submission contain map images which may be copyrighted. All PLOS content is published under the Creative Commons Attribution License (CC BY 4.0), which means that the manuscript, images, and Supporting Information files will be freely available online, and any third party is permitted to access, download, copy, distribute, and use these materials in any way, even commercially, with proper attribution. For these reasons, we cannot publish previously copyrighted maps or satellite images created using proprietary data, such as Google software (Google Maps, Street View, and Earth). For more information, see our copyright guidelines: http://journals.plos.org/plosone/s/licenses-and-copyright.

1) You may seek permission from the original copyright holder of Figure 2 to publish the content specifically under the CC BY 4.0 license.  

2) If you are unable to obtain permission from the original copyright holder to publish these figures under the CC BY 4.0 license or if the copyright holder’s requirements are incompatible with the CC BY 4.0 license, please either i) remove the figure or ii) supply a replacement figure that complies with the CC BY 4.0 license. Please check copyright information on all replacement figures and update the figure caption with source information. If applicable, please specify in the figure caption text when a figure is similar but not identical to the original image and is therefore for illustrative purposes only.

**Additional Editor Comments:**

- Please disregard the comments from reviewer 1.

Reviewers' comments:

Reviewer's Responses to Questions

**Comments to the Author**

1. Is the manuscript technically sound, and do the data support the conclusions?

Reviewer #1: Yes

Reviewer #2: Yes

Reviewer #3: Yes

2. Has the statistical analysis been performed appropriately and rigorously? 

Reviewer #1: Yes

Reviewer #2: Yes

Reviewer #3: Yes

3. Have the authors made all data underlying the findings in their manuscript fully available?

Reviewer #1: Yes

Reviewer #2: Yes

Reviewer #3: No

4. Is the manuscript presented in an intelligible fashion and written in standard English?

Reviewer #1: Yes

Reviewer #2: Yes

Reviewer #3: Yes

5. Review Comments to the Author

Reviewer #1: This study aimed to assess whether there were changes in eating habits during the 35

restriction measures due to the Covid-19 pandemic. This is an interesting manuscript regarding the statistical analysis used. However, the time they evaluated changes in dietary due to the pandemic was too short (only four months of social restriction), and the results were expected.

Materials and Methods

Line 83: Why did you use the prevalence of 50% to change dietary patterns? Is there a study relating to this? I suggest citing the reference you used.

Line 85: How did you invite the people to participate in your study? Did you share the invite on social media or with a specific public?

Line 118-125: This paragraph should be in the results, presenting the results of Table 1.

Results

Table 2 should be before Table 1, presenting the characterization of the sample.

Include the number of participants in the title tables.

Table 4: include in the footnote that the low adherence was the reference for the analysis.

Discussion

The discussion is too long and does not mention the short period analyzed before the start of the social restriction. I recommend evaluating this manuscript to publish as a short communication. This way, you could rewrite your discussion and make it more concise.

Reviewer #2: Commentary on the study “Estimation of the future prevalence of diabetes based on data from the Brazilian Study of Cardiovascular Risk Factors in Adolescents (ERICA)”

Overview:

This study was a secondary analysis from a multicenter study of Adolescents. The title is interesting; however, I have some comments.

1. In the abstraction section, please explain more the objective of this study.

2. Line 53 and 54, please add references for this sentence.

3. "Diabetes-related costs in Brazil were calculated at BRL 52.3 billion in 2019…" before this sentence you can mention in one sentence that " Various factors such as genetics, nutrition factors, inactivity and other environmental factors are involved in diabetes etiology."you can read and add these articles as references to this sentence: "Association of circulating adipokines with metabolic dyslipidemia in obese versus non-obese individuals" and " Effect of Calcium and Vitamin D Co-supplementation on Blood Pressure: A Systematic Review and Meta-Analysis"

4. In the introduction section and paragraph 3, the transition from the global impact of diabetes to the objectives of the study is somewhat abrupt. It would benefit from a clearer lead-in statement, outlining the purpose of this particular research in the context of existing studies.

5. it is not clear why specific predictive models from the literature were chosen over others. Expanding on the justification for the model selection could strengthen the rigor of the method.

6. What was the inclusion criteria to choose the studies?

7. In term of logistic regression, please more explain the selected variables

8. Provide more detail on validation processes and acknowledge the limitations of the chosen model, including confounding factors

9. In the result section, its better to explain more about sensitivity analyses. For example, the results indicate a wide range (1.1% to 28%) in the sensitivity analyses. The implications of this broad range for public health policy and intervention strategies could be discussed more thoroughly.

Reviewer #3: The paper presents a valuable analysis of predicted prevalence of type 2 diabetes mellitus (T2DM) based in a representative national sample of Brazilian adolescents. On top of that, the study explores the most relevant cardiovascular risk factors related to the development of T2DM that are important to subside health policies aiming to prevent this disease and promote the improvement of health conditions.

Minor comments:

Introduction (lines 117-118): The study aim is to analyse the prevalence of T2DM in the ERICA population or use ERICA´s data to predict the future prevalence of diabetes? Please review the aim of the study to make it clear for the readers.

Table 7: I suggested changing the name of the column “Individuals with T2DM by state” to “Estimated number of individuals with T2DM” to avoid confusion. The current name of the column make it think that this is the actual number of individuals with diabetes in the ERICA study data. Additionally, it is not necessary to say that is by state because the table is already showing the values by each state.

Results (Line 340): It is not clear how the period of 20.5 years were considered in the model. Is this a parameter in the model?

Discussion: The authors considered only biological factors in the predictive model of T2DM, but it is already known that social factors are also important in the development and progression of this disease. The authors could also include this topic on the discussion when describing the importance of having a predictive model developed to the Brazilian population. This model could then consider the socioeconomic factors (i.e. income, education, housing) to better predict the future occurrence of type 2 diabetes.

6. PLOS authors have the option to publish the peer review history of their article (what does this mean? ). If published, this will include your full peer review and any attached files.

**Do you want your identity to be public for this peer review?** For information about this choice, including consent withdrawal, please see our Privacy Policy .

Reviewer #1: No

Reviewer #2: No

Reviewer #3: No

---

## [Author Response · Author response to Decision Letter 1]

23 Apr 2025

Dear Editor and Reviewers,

We have carefully reviewed the formatting of our manuscript to ensure compliance with PLOS ONE's style guidelines, and it appears to be in accordance with the provided templates.

Regarding data availability, our study utilized third-party data sources, including the ERICA Study, Vigitel, and the Bogalusa Heart Study. As we do not own these datasets, we are unable to publicly share the raw data. However, we have updated the manuscript to clarify the process for accessing these datasets within the Data Availability Statement section, where we provide the necessary information for researchers to request access through the respective data providers. Additionally, Figure 2 was created entirely by the authors. To ensure compliance with CC BY 4.0 licensing, we have included the following statement in its figure legend: "This figure was created by the authors and is published under the Creative Commons Attribution 4.0 International (CC BY 4.0) license." These revisions ensure that our manuscript meets PLOS ONE’s formatting, data accessibility, and copyright requirements.

Moreover, we sincerely appreciate the time and effort the reviewers and editorial team have dedicated to evaluating our manuscript. Their constructive feedback has been invaluable in improving the clarity, methodological rigor, and overall quality of our study. On the Response to reviewers, we provide a point-by-point response to the comments raised by Reviewers 2 and 3. We have carefully addressed each suggestion and made the necessary modifications to the manuscript.

Regards

---

## [Decision Letter · Decision Letter 1]

30 May 2025

Estimation of the future prevalence of diabetes based on data from the Brazilian Study of Cardiovascular Risk Factors in Adolescents (ERICA).

PONE-D-24-44409R1

Dear Dr. Pozzi Ottavio,

We’re pleased to inform you that your manuscript has been judged scientifically suitable for publication and will be formally accepted for publication once it meets all outstanding technical requirements.

Kind regards,

Hidetaka Hamasaki

Academic Editor

PLOS ONE

Additional Editor Comments (optional):

Reviewers' comments:

Reviewer's Responses to Questions

**Comments to the Author**

1. If the authors have adequately addressed your comments raised in a previous round of review and you feel that this manuscript is now acceptable for publication, you may indicate that here to bypass the “Comments to the Author” section, enter your conflict of interest statement in the “Confidential to Editor” section, and submit your "Accept" recommendation.

Reviewer #2: All comments have been addressed

Reviewer #3: All comments have been addressed

2. Is the manuscript technically sound, and do the data support the conclusions?

Reviewer #2: Yes

Reviewer #3: (No Response)

3. Has the statistical analysis been performed appropriately and rigorously? 

Reviewer #2: Yes

Reviewer #3: (No Response)

4. Have the authors made all data underlying the findings in their manuscript fully available?

Reviewer #2: Yes

Reviewer #3: (No Response)

5. Is the manuscript presented in an intelligible fashion and written in standard English?

Reviewer #2: Yes

Reviewer #3: (No Response)

6. Review Comments to the Author

Reviewer #2: Authors could answer all of my comments in the previous version of this manuscript, so I do not have additional comments.

Reviewer #3: (No Response)

7. PLOS authors have the option to publish the peer review history of their article (what does this mean? ). If published, this will include your full peer review and any attached files.

**Do you want your identity to be public for this peer review?** For information about this choice, including consent withdrawal, please see our Privacy Policy .

Reviewer #2: **Yes: ** Mehran Rahimlou

Reviewer #3: No

---

## [Editor Report · Acceptance letter]

PONE-D-24-44409R1

PLOS ONE

Dear Dr. Pozzi Ottavio,

I'm pleased to inform you that your manuscript has been deemed suitable for publication in PLOS ONE. Congratulations! Your manuscript is now being handed over to our production team.

Kind regards,

on behalf of

Dr. Hidetaka Hamasaki

Academic Editor

PLOS ONE